# Learning User Perceived Clusters with Feature-Level Supervision

Ting-Yu Cheng   Kuan-Hua Lin   Xinyang Gong   Kang-Jun Liu   Shan-Hung Wu
{tycheng,khlin,xygong,kjliu}@datalab.cs.nthu.edu.tw   shwu@cs.nthu.edu.tw

## Abstract

Semi-supervised clustering algorithms have been proposed to identify data clusters that align with user perceived ones via the aid of side information such as seeds or pairwise constrains. However, traditional side information is mostly at the *instance level* and subject to the sampling bias, where non-randomly sampled instances in the supervision can mislead the algorithms to wrong clusters. In this paper, we propose learning from the *feature-level* supervision. We show that this kind of supervision can be easily obtained in the form of *perception vectors* in many applications. Then we present novel algorithms, called *Perception Embedded* (PE) clustering, that exploit the perception vectors as well as traditional side information to find clusters perceived by the user. Extensive experiments are conducted on real datasets and the results demonstrate the effectiveness of PE empirically.

## 1   Introduction

Data clustering [21], one of the most important unsupervised learning tasks, has found numerous applications in different domains, including computer vision [14], information retrieval [4, 20], recommender systems [17], etc. Given a dataset, the goal is to find groups of instances, called *clusters*, such that instances in the same cluster are more "similar" (based on a certain similarity metric) to each other than to those in other clusters. We call the clusters found by a clustering algorithm the *data-driven clusters*.

In practice, different users may look the same dataset differently and may cluster instances in their own ways. We call such clusters the *user-perceived clusters*. Due to its unsupervised nature, data clustering is an ill-posed problem [16] that may not be able to capture the variety of user perception. Semi-supervised clustering [28] is proposed to address this limitation by allowing individual users to provide side information about their perceived clusters. The side information can have different forms, such as the *seeds*, i.e., instances that must belong to certain clusters, or the pairwise *must-link/cannot-link constraints* respectively ensuring/preventing two instances to go/from going to the same cluster. Most of the existing studies on semi-supervised clustering focus on finding clusters that align with the side information, by either modifying the clustering algorithms [28, 18, 29, 19, 10, 30, 23, 15, 25] or similarity metric [13, 3, 2, 27, 31, 8, 5, 33]. However, few studies have investigated how close are the data-driven clusters to the user-perceived clusters in real situations.

In this paper, we conduct experiments on Amazon Mechanical Turk platform to obtain the user-perceived clusters, compare them with the data-driven clusters returned by existing semi-supervised clustering algorithms, and observe a large gap—in many cases, only about half of the instances in a user-perceived cluster can be successfully identified by the closest data-driven cluster. This is mainly due to the *sampling bias*: with limited resources (e.g., time, budget, etc.), the seeds/constraints provided by a user usually apply to only a small portion of instances in the dataset. These instances, when locating at ambiguous positions (in the feature space) such as the cluster boundaries or overlapping areas, can mislead the semi-supervised clustering algorithms to results that align poorly with actual user perception. This can be illustrated by Figures 1(a)-(c). The performance of existing

semi-supervised clustering algorithms depends largely on the sampling of the seeded/constrained instances.

Unfortunately, the sampling of seeded/constrained instances may be hard to control due to many practical reasons. For example, when recommending products to a user in an e-commerce system based on the product clusters seeded by those products viewed/rated/liked by the user, the seeds are unlikely to be uniformly distributed within each user-perceived cluster because the user may choose to interact with the products following a particular order such as the price or bestselling scores, etc. Similar difficulties can be identified in other applications, and get even worse in big or high-dimensional datasets known to be multifaceted. It is crucial to bridge the current gap between user-perceived and data-driven clusters.

Observing that a user usually provides the seeds/constraints based on certain reasons he/she perceives, we assume these reasons can be captured as another side information called *perception vectors*, and study the following problem:

**Problem 1.** Given a dataset $\mathcal{X} = \{\boldsymbol{x}_i\}_{i=1}^n$ of instances $\boldsymbol{x}_i \in \mathbb{R}^d$, an integer $k$, and some side information (e.g., the seeds $\mathcal{S} = \{\mathcal{S}_j\}_{j=1}^k$, $\mathcal{S}_j \subset \mathcal{X}$ or pairwise must-link $\mathcal{L}_{must} = \{(i,j)\}_{i \neq j}$ and cannot-link $\mathcal{L}_{not} = \{(i,j)\}_{i \neq j}$ constraints, $1 \leq i, j \leq n$) along with the associated perception vectors $\mathcal{P} = \{\boldsymbol{p}_j\}_{j=1}^{|\mathcal{S}| \text{ or } |\mathcal{L}_{must}|+|\mathcal{L}_{must}|}$, $\boldsymbol{p}_j \in \mathbb{R}^b$ or null, provided by a user that partially reveal the unknown user-perceived clusters $\mathcal{D} = \{D_j\}_{j=1}^k$, $D_j \subset \mathcal{X}$. Find $k$ clusters $\mathcal{C} = \{C_j\}_{j=1}^k$, $C_j \subset \mathcal{X}$, aligning with $\mathcal{D}$ most.

Note that a user need *not* know the entire $\mathcal{D}$ in advance to provide the perception vectors $\mathcal{P}$, as $\mathcal{P}$ is only associated with the seeds/constraints. Also, a seed or pairwise constraint need *not* have a perception vector, if the user experience difficulty in providing the latter. Furthermore, the features of instances and those of perception vectors need *not* lie in the same space. For example, when clustering images, the instance features may be the RGB values while the perception features may be the names of objects in images. However, we argue that the perception features are usually correlated to the instance features and can be helpful in finding better alignment between $\mathcal{C}$ and the unknown $\mathcal{D}$.

The perception vectors can be easily obtained in many practical cases, either explicitly or implicitly. For example, whenever a user $i$ is giving a seed set $\mathcal{S}_j$, we can ask the user to write down explicitly the reasons behind. In this case, $\boldsymbol{p}_j$ can be the bags of the written words. When clustering musics/videos for making recommendations to a user, one may extract $\boldsymbol{p}_j$ implicitly from the title/description of a YouTube playlist organized by the user. Similarly, the $\boldsymbol{p}_j$ can also be extracted implicitly from the name/description of a Facebook group (when clustering people), or even search keywords associated with a click stream (when clustering web pages). For more discussions about the availability and extraction of perception vectors, please refer to Section 1 of the supplementary materials.

To solve Problem 1, we generalize the spectral embedded clustering [12] (which covers popular $k$-means variants) and propose a family of new algorithms, called the *Perception Embedded* (PE) clustering that allows the perception vectors to act as the *feature-level supervision* complementing traditional *instance-level supervision* (i.e., seeds/constraints), which are subject to the biased instance sampling. Extensive experiments are conducted and the results show that PE significantly reduces the occurrence of "bad" data-driven clusters due to sampling bias and can lead to up to 63% improvement in $F$-score as compared to user-perceived clusters in the ground truth.

**Further Related Work.** Poon et al. [24] propose an active framework that iteratively asks the user to select a subset of instance features to mitigate the impact of initial sampling of supervised instances. But this framework is infeasible to applications where the questions (and their sampling) are not

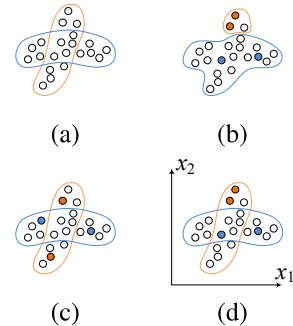

(a)         (b)

(c)         (d)

Figure 1: Sampling bias. (a) User-perceived clusters. (b) Data-driven clusters with misleading seeds. (c) Data-driven clusters with representative seeds. (d) Data-driven clusters found by PE with misleading seeds and perception vectors. Through the novel embedding, PE knows that the first data feature ($x_1$) *cannot* explain the blue seeds and only propagates the blue cluster labels to other instances that look similar in the second data feature ($x_2$). Similarly, PE propagates the orange cluster labels only along the first data feature.

allowed. Being orthogonal and compatible to this approach, PE asks the user just once (to provide $\mathcal{P}$ and other side information) and learns from the "best" instance features automatically via a better embedding guided by $\mathcal{P}$.

## 2   Perception Gap

In this section, we investigate how users perceive the clusters differently from existing semi-supervised clustering algorithms. To collect the user-perceived clusters *directly from users*, we conduct a series of experiments on the Amazon Mechanical Turk platform using an image datatset $\mathcal{X}$, called the *Mturk image dataset*. In each experiment $e$, a human worker is asked to cluster the images in $\mathcal{X}$ based on his/her own opinion, as shown in Figure 2, to provide the user-perceived clusters $\mathcal{D}^{(e)}$ as well as the perception vectors $\mathcal{P}^{(e)}$. To make the experiments manageable for human workers, we create $\mathcal{X}$ by randomly selecting 180 images from the NUS-WIDE [6] image database. Each image $x_i \in \mathcal{X}$ has 500 features consisting of the bag of words based on SIFT descriptions. We control the quality of data collection by 1) making tutorials that encourage the workers to provide independent clusters; 2) skipping spoilers; 3) eliminating low-quality clusters; and 4) carefully preprocessing the raw perception features, including stop-word elimination and stemming, etc.. Finally, we get

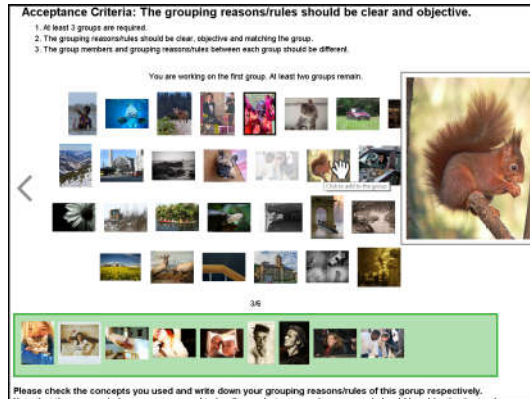

Figure 2: Screenshot of an experiment, where a worker is asked to 1) identify at least 3 image groups by dragging the randomly ordered pictures into different green boxes (each forming a user-perceived cluster $D_j$) with replacement; and 2) write down some words about each identified group (forming a perception vector $p_j$).

325 user-perceived clusters from 100 users (whose accept rates are greater or equal to 95%) and their corresponding perception vectors consisting of 108 features. Statistics of the ground truth clusters $\mathcal{D}^{(e)}$ are shown in Table 1 in supplementary. In average, each perception vector has about 2 words (i.e., non-zero feature values), which is sparse (among 108).

We study the difference between the user-perceived and data-driven clusters when the latter are found by the semi-supervised clustering algorithms with seeds. For each set $\mathcal{D}^{(e)}$ of clusters perceived by a user from experiment $e$, we calculate $F(\mathcal{C}, \mathcal{D}^{(e)}) \in [0, 1]$, an $F$-score indicating how similar the clusters in $\mathcal{C}$ and $\mathcal{D}^{(e)}$ are. The precise definition of $F(\mathcal{C}, \mathcal{D}^{(e)})$ is given in Section 4. For each cluster $D_j^{(e)} \in \mathcal{D}^{(e)}$, we randomly pick two instances as seeds and move them to the seed set $S_j^{(e,s)}$, where $s$ denotes a particular choice of the seeds. We then modify the Overlapping $k$-Means (OKM) algorithm [7][1] to a seeded version, denoted as OKM*, by following the technique proposed by [28], and use it along with $S_j^{(e,s)}$ to find the data-driven clusters $\mathcal{C}^{(e,s)}$. Figure 3(a) shows the histogram of $\frac{1}{s}\sum_s F(\mathcal{C}^{(e,s)}, \mathcal{D}^{(e)})$'s over all $e$'s. We can see that OKM* with side information (light blue bars) does improve the overall performance as compared to OKM (dark blue bars), by increasing the average $F$-score (across different users) from 0.4 to 0.5. However, when looking into the histogram of $F(\mathcal{C}^{(e,s)}, \mathcal{D}^{(e)})$'s over a randomly picked $e$ and all $s$'s , as shown in Figure 3(b) (blue bars),[2] we find that there still exists a fair amount of seed sets that lead to $F$-score as low as 0.3. Some seeds are clearly not helpful, justifying the existence of sampling bias. We observe similar results for different clustering algorithms and different experiments $e$'s.[3] Clearly, there is still a large gap between what users want and what existing semi-supervised clustering algorithms can provide.

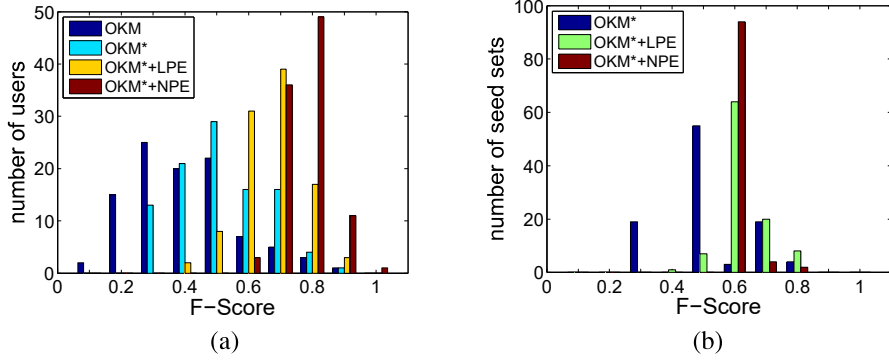

(a)                                             (b)

Figure 3: (a) Histogram of $F$-scores when comparing $\mathcal{C}$ and $\mathcal{D}$'s of different users, where $\mathcal{C}$ is found using a clustering algorithm. (b) Histogram of $F$-scores when comparing $\mathcal{C}$'s and $\mathcal{D}$ of a randomly picked user, where $\mathcal{C}$'s are found using a seeded clustering algorithm given different seed sets $\mathcal{S}$'s.

## 3   PE Clustering

In this section, we introduce the *Perception Embedded* (PE) clustering to solve Problem 1. The goal is to turn the perception vectors given by a user into the *feature-level supervision* that identifies the data-features which the user really cares about. This new kind of supervision complements existing *instance-level supervision* (i.e., seeds/constraints) vulnerable to uncontrollable instance sampling. The above goal is challenging because the perception features and data features may *not* be linearly correlated. For example, a (binary) perception feature "car" is likely to be nonlinearly correlated with the (numeric) RGB features of an image. We propose two specialized PE algorithms, named *Linear* PE (LPE) and *Nonlinear* PE (NPE), to deal with the linear and nonlinear correlations respectively.

### 3.1   Linear PE (LPE)

The basic idea of LPE is to embed the space of data features into that of the perception features, so the perception features can guide the embedding. This gives the following regularized objective:

$$\arg\min_{\boldsymbol{F}\geq\boldsymbol{O},\boldsymbol{W}\geq\boldsymbol{O},b\geq\boldsymbol{0}} \|\boldsymbol{X}\boldsymbol{W}+\boldsymbol{1}_n\boldsymbol{b}^\top-\boldsymbol{F}\|_F^2+\beta\,\|\boldsymbol{S}(\boldsymbol{F}-\boldsymbol{P})\|_F^2\,, \tag{1}$$

where $\boldsymbol{X}=[\boldsymbol{x}_1,\cdots,\boldsymbol{x}_n]^\top\in\mathbb{R}^{n\times d}$ is the data feature matrix, $\boldsymbol{F}=[\boldsymbol{f}_1,\cdots,\boldsymbol{f}_n]^\top\in\mathbb{R}^{n\times b}$ is a latent representation matrix whose each row $\boldsymbol{f}_i\in\mathbb{R}^b$ denotes the latent representation of the instance $i$. The (space of) $\boldsymbol{X}$ is embedded into (that of) $\boldsymbol{F}$ via a linear embedding parametrized by $\boldsymbol{W}\in\mathbb{R}^{d\times b}$ and $\boldsymbol{b}\in\mathbb{R}^b$, which respectively denote the weight matrix and intercept vector. The $\boldsymbol{1}_n\in\mathbb{R}^n$ is a vector of all 1's and $\|\cdot\|_F$ denotes the Frobenius norm. The $\boldsymbol{P}\in\mathbb{R}^{n\times b}$ is the perception matrix whose each row $\boldsymbol{p}_i$ denotes the perception vector associated with the instance $i$, and $\boldsymbol{S}\in\mathbb{R}^{n\times n}$ is a diagonal matrix whose each entry $\boldsymbol{S}_{i,i}$ equals 1 if the $i$-th instance is a seed or pairwise constrained and 0 otherwise, and $\beta$ is a hyperparameter.

The first term of Eq. (1) is identical to the loss term of Spectral Embedded Clustering (SEC) [12], which is known to be a generalization of the discriminative $k$-means clustering. To model the instance-level supervision like pairwise constraints, we add a term that penalizes the dissimilarity of latent representation of instances having must-link constraints, as below:

$$\arg\min_{\boldsymbol{F}\geq\boldsymbol{O},\boldsymbol{W}\geq\boldsymbol{O},b\geq\boldsymbol{0}} \|\boldsymbol{X}\boldsymbol{W}+\boldsymbol{1}_n\boldsymbol{b}^\top-\boldsymbol{F}\|_F^2+\alpha\|\boldsymbol{M}\boldsymbol{F}\|_F^2+\beta\,\|\boldsymbol{S}(\boldsymbol{F}-\boldsymbol{P})\|_F^2\,, \tag{2}$$

where $\boldsymbol{M}\in\mathbb{R}^{n^2\times n}$ is a pairwise comparison matrix, whose each row $\boldsymbol{M}_{(s,t),:}$ corresponds to some pair $(s,t)\in\mathcal{L}_{must}$ and $\boldsymbol{M}_{(s,t),i}=1$ if $i=s$, $-1$ if $i=t$, and 0 otherwise, and $\alpha$ is a hyperparameter.

---

experiment, we replace an image back to $\mathcal{X}$ each time after it is put into a user-perceived cluster. So the user can decide whether it should be included in another user-perceived cluster.

[2]In this experiment, the user from the picked experiment identifies 3 clusters with size 4, 6, 4 respectively, so there are $\binom{4}{2}\times\binom{6}{2}\times\binom{4}{2}=540$ seed sets.

[3]For more details, please refer to Section 4.2 of the supplementary materials.

To model the seed sets $\mathcal{S} = \{\mathcal{S}_l\}_{l=1}^k$, we can simply transform them into the pairwise must-link constraints $\mathcal{L}_{must} = \{(i,j) : i \neq j \text{ and } (\boldsymbol{x}_i, \boldsymbol{x}_j) \in \mathcal{S}_l \text{ for some } l\}$.

**Further Refinement.** We regularize the norm of $\boldsymbol{W}$ in Eq. (2) to prefer simpler embedding that is more robust to noise. Also, we discard the second term as its effect can be covered by the third term. The details are left to Section 2 of the supplementary to avoid distraction. The final LPE objective becomes:

$$\arg\min_{\boldsymbol{F} \geq \boldsymbol{O}, \boldsymbol{W} \geq \boldsymbol{O}, \boldsymbol{b} \geq \boldsymbol{0}} \left\| \boldsymbol{X}\boldsymbol{W} + \boldsymbol{1}_n \boldsymbol{b}^\top - \boldsymbol{F} \right\|_F^2 + \beta \left\| \boldsymbol{S}(\boldsymbol{F} - \boldsymbol{P}) \right\|_F^2 + \gamma \left\| \boldsymbol{W} \right\|_F^2 \tag{3}$$

Note that the entries of $\boldsymbol{F}$, $\boldsymbol{W}$, and $\boldsymbol{b}$ need to be non-negative to preserve the clustering interpretation [11]. With these non-negative constrains, the final LPE objective is hard to solve using traditional first- or second-order methods. We will describe our training steps later.

## 3.2 Nonlinear PE (NPE)

When the perception features are nonlinearly correlated with data features, the above linear embedding may be too simple to capture the correlation and identify the data features that the user really cares about. Next, we show that LPE can be readily kernelized to achieve nonlinear embedding.

We first map each instance $\boldsymbol{x}$ to a function $\Phi(\boldsymbol{x})$, $\Phi(\boldsymbol{x})(\cdot) = k(\boldsymbol{x}, \cdot)$ where $k$ is a given kernel function, in a high dimensional Reproducing Kernel Hilbert Space (RKHS) $\mathcal{H}$. Consider a function $w_j \in \mathcal{H}$ that embeds the instances to a single perception feature $j$. We have $w_j(\boldsymbol{x}) = \langle w_j, \Phi(\boldsymbol{x}) \rangle \in \mathbb{R}$ due to the reproducing property of an RKHS. So, the embedding is nonlinear if the kernel function $k$ is nonlinear (e.g., Gaussian RBF kernel). Following the concept of perception embedding behind Eq. (3), we want $w_j(\boldsymbol{x}_i)$ to be as close to the true perception feature $p_{i,j}$ as possible (if $p_{i,j}$ is presented). We let $w_j$ be the minimizer of the following regularized functional:

$$\arg\min_{\boldsymbol{F}_{:,j} \geq \boldsymbol{0}, w_j \in \mathcal{H}} \sum_{i=1}^n (w_j(\boldsymbol{x}_i) - \boldsymbol{F}_{i,j})^2 + \beta \sum_{i=1}^n (\boldsymbol{S}_{i,i}(\boldsymbol{F}_{i,j} - \boldsymbol{P}_{i,j}))^2 + \gamma \left\| w_j \right\|_{RKHS}^2. \tag{4}$$

**Theorem 2.** *Given any $\boldsymbol{F}_{:,j} \geq \boldsymbol{0}$, the solution $w_j$ of Eq. (4) admits the form $w_j = \sum_{i=1}^n c_{i,j}\Phi(\boldsymbol{x}_i)$.*

**Proof:** Observe that any $w_j \in \mathcal{H}$ can be represented as $w_j(\cdot) = \sum_{i=1}^n c_{i,j}k(\boldsymbol{x}_i, \cdot) + w_\perp(\cdot)$, where $k(\boldsymbol{x}_i, \cdot) = \Phi(\boldsymbol{x}_i)$ and $w_\perp \in \mathcal{H}$ is orthogonal to $k(\boldsymbol{x}_i, \cdot)$ for $i = 1, 2, \cdots, n$. By the Reproducing properties and $\langle w_\perp, k(\boldsymbol{x}_i, \cdot) \rangle = 0$ for any $i$, we have

$$w_j(\boldsymbol{x}_i) = \langle w_j, k(\boldsymbol{x}_i, \cdot) \rangle = \sum_{l=1}^n c_{l,j}k(\boldsymbol{x}_i, \boldsymbol{x}_l) + \langle w_\perp, k(\boldsymbol{x}_i, \cdot) \rangle = \sum_{l=1}^n c_{l,j}k(\boldsymbol{x}_i, \boldsymbol{x}_l). \tag{5}$$

Next, let $L_{w_j}(\boldsymbol{x}_i) = \sum_{i=1}^n (w_j(\boldsymbol{x}_i) - F_{i,j})^2 + \beta \sum_{i=1}^n (S_{i,i}(F_{i,j} - P_{i,j}))^2$. Suppose $w_j^* \in \mathcal{H}$ is the minimizer of $L_{w_j}(\boldsymbol{x}_i) + \gamma \left\| w_j \right\|_{RKHS}^2$ and has the form $w_j^*(\cdot) = \sum_{i=1}^n c_{i,j}^* k(\boldsymbol{x}_i, \cdot) + w_\perp^*(\cdot)$. We show that the function $w_j^\ddagger(\cdot) = \sum_{i=1}^n c_{i,j}^* k(\boldsymbol{x}_i, \cdot)$ will always be a better solution, a contradiction. Based on Eq. (5) we have $w_j^*(\boldsymbol{x}_i) = \sum_{l=1}^n c_{l,j}^* k(\boldsymbol{x}_l, \boldsymbol{x}_i) + w_\perp^*(\boldsymbol{x}_i) = \sum_{l=1}^n c_{l,j}^* k(\boldsymbol{x}_l, \boldsymbol{x}_i) = w_j^\ddagger(\boldsymbol{x}_i)$, implying $L_{w_j^*}(\boldsymbol{x}_i) = L_{w_j^\ddagger}(\boldsymbol{x}_i)$. Then we have

$$\begin{aligned} L_{w_j^\ddagger}(\boldsymbol{x}_i) + \gamma \parallel w_j^\ddagger \parallel_{RKHS}^2 &= L_{w_j^*}(\boldsymbol{x}_i) + \gamma \sum_{i,l=1}^n c_{i,j}^* c_{l,j}^* k(\boldsymbol{x}_i, \boldsymbol{x}_l) \\ &\leq L_{w_j^*}(\boldsymbol{x}_i) + \gamma \sum_{i,l=1}^n c_{i,j}^* c_{l,j}^* k(\boldsymbol{x}_i, \boldsymbol{x}_l) + \gamma \parallel w_\perp^* \parallel_{RKHS}^2 = L_{w_j^*}(\boldsymbol{x}_i) + \gamma \parallel w_j^* \parallel_{RKHS}^2. \end{aligned}$$

That is, the objective score of Eq. (4) given by $w_j^\ddagger$ is always smaller than that given by $w_j^*$. $\qquad\square$

The above theorem shows that instead of $w_j$, we can find $c_{i,j}$'s for all $i$ (and $\boldsymbol{F}_{:,j}$) to solve Eq. (4). Now consider embedding instances to all $b$ perception features via $w_1, w_2, \cdots, w_b$. We find $c_{i,j}$'s for all $i$ and $j$. This gives the NPE objective:

$$\arg\min_{\boldsymbol{F} \geq \boldsymbol{O}, \boldsymbol{C}} \left\| \boldsymbol{K}\boldsymbol{C} - \boldsymbol{F} \right\|_F^2 + \beta \left\| \boldsymbol{S}(\boldsymbol{F} - \boldsymbol{P}) \right\|_F^2 + \gamma \operatorname{tr}(\boldsymbol{C}^\mathsf{T}\boldsymbol{K}\boldsymbol{C}), \tag{6}$$

where $\boldsymbol{K} \in \mathbb{R}^{n \times n}$ is the kernel matrix and $\boldsymbol{C} = [\boldsymbol{c}_1, \cdots, \boldsymbol{c}_b] \in \mathbb{R}^{n \times b}$ whose each column $\boldsymbol{c}_j = [c_{1,j}, c_{2,j}, \cdots, c_{n,j}]^\top \in \mathbb{R}^n$. Again, the non-negative constraint of $\boldsymbol{F}$ makes this objective hard to solve using traditional methods.

### 3.3 Training

To solve the non-negative $F$, $b$ and $W$ in LPE, we followed the multiplicative update rules proposed by Long et al. [22] to update $F$, $b$ and $W$ iteratively. To solve the non-negative $F$ and $C$ in NPE, we first observe that both $K$ and $F$ have non-negative elements. So, to be efficient, we narrow the search space of $C$ down to the positive matrices. We then apply the multiplicative update rules [22] to update $F$ and $C$ iteratively. Due to the space limitation, the update rules and algorithms are left to Section 3 of the supplementary.

## 4 Performance Evaluation

**Baselines and Settings.** We compare PE with other semi-supervised clustering algorithms that support seeds. Recall that in Section 2, we find that users tend to perceive overlapping clusters, so the data-driven clusters should be overlapping as well. We consider clustering algorithms that 1) accept seeds and 2) deal with overlapping clusters. However, few algorithms achieve these two goals off-the-shelf. We therefore modify existing seeded/overlapping clustering methods when necessary. We name the modified version of a clustering algorithm by appending a star "*" to its original name. We first consider the seeded versions of overlapping clustering algorithms. We modify the *Overlapping k-Means* (OKM) [7] and and Model-based Overlapping Clustering (MOC) [1] by using the technique proposed in [28] so they take the seeds as input. On the other hand, we consider the overlapping versions of semi-supervised clustering algorithms. We extend the seeded Spectral Embedded Clustering (SEC) [12] and the Constrained Clustering by Spectral Kernel Learning (CCSKL) [18] by employing the OKM (instead of the ordinary $k$-means) in their last step so they can produce overlapping clusters. Note that we also consider other semi-supervised clustering algorithms [9, 32] but it turns out that they have convergence problems and/or perform poorly in our experiments, as their models assume a relatively large portion of seed/constraints among the dataset. This assumption does *not* hold in many real-world applications as a user often have limited resources (e.g., time) thus could only annotate a tiny portion of the possibly huge dataset (otherwise, the user could have just clustered the dataset manually to get perfect user-perceived clusters). So, we set the portion of seeds relatively small (specifically, each cluster contains only 2 seeds or one must-link constraint, making the problem very challenging) and do not show the results of these studies [9, 32]. We adopt the Gaussian RBF kernel for NPE. For detailed setups such as hyperparameter tuning, please see Section 4.1 of the supplementary.

**Evaluation Metrics.** Both LPE and NPE can pair up with each of the above baselines to output final clusters. We compare the performance of the baselines with the pairings. To evaluate the quality of the founded clusters $\mathcal{C}$. We compare them with the user-perceived clusters $\mathcal{D}$ using the $F$-score: $F(\mathcal{C}, \mathcal{D}) = \frac{2 \cdot \text{precision}(\mathcal{C},\mathcal{D}) \cdot \text{recall}(\mathcal{C},\mathcal{D})}{\text{precision}(\mathcal{C},\mathcal{D}) + \text{recall}(\mathcal{C},\mathcal{D})} \in [0, 1]$. We define $\text{recall}(\mathcal{C}, \mathcal{D}) = \frac{1}{k} \sum_{j=1}^{k} |C_{j^*} \cap D_j| / |D_j|$ and $\text{precision}(\mathcal{C}, \mathcal{D}) = \frac{1}{k} \sum_{j=1}^{k} |C_{j^*} \cap D_j| / |C_{j^*} \cap \mathcal{D}|$,[4] where $C_{j^*} \in \mathcal{C}$ denotes a data-driven cluster with the best index $j^*$ maximizing recall or precision (we try out all possible permutation of the clusters in $\mathcal{C}$ to determine the best index $j^*$). The higher $F(\mathcal{C}, \mathcal{D})$, the more similar between clusters in $\mathcal{C}$ and $\mathcal{D}$.

**Datasets.** In addition to the Mturk image dataset, we consider the *song* [26] and *citation* (Citation Prediction Task of KDD Cup 2003) datasets as well in our experiments. **Song Dataset.** The song dataset is adapted from the MusiClef 2012 Multimodal Music dataset [26], which contains 1355 music songs. The song dataset contains the low-level audio features, block-level features and PS09 features (please refer to the origin paper [26]), and high-level tags on the songs. The songs are tagged by professional music annotators with respect to genre and mood aspects. There are totally 94 distinct tags such as "acoustic", "punk", and "slow", etc. Each song has multiple tags. We simulate users and their clusters as follows. There are 10 users. Each user has 3 clusters. And each cluster has 2 randomly assigned tags as perception features. The songs that have the corresponding tags belong to the corresponding cluster. So, among all tags, we randomly select 20 pairs of tags as the perception features. Meanwhile, we ensure that size of each cluster is greater than 10. Note that there

are totally 30 clusters, but there are only 20 pairs of tags. This means that some clusters have the same perception features. We randomly assign clusters to users while ensuring that clusters of the same user do not have the same perceptions and the cluster combinations are distinct among all the users. Finally, we get 30 clusters from 10 users, 530 songs, 10,783 data features, and 26 perception features. **Citation Dataset.** The citation dataset is adapted from the Citation Prediction Task of KDD Cup 2003. This dataset contains the LaTeX sources of about 29,000 papers in the hep-th portion of the arXiv until May 1, 2003. We parse papers in 2003, which are expected to cite more arXiv hep-th papers in the past, and form the user-perceived clusters from the *citation groups*—sets of arXiv hep-th papers that are cited together (i.e., bracketed in the same tex citation syntax) by other papers. Each citation group in a paper is viewed as a user-perceived cluster identified by the authors of that paper. To avoid extreme situations, we exclude the citation groups that contain less than 2 arXiv hep-th papers and the papers that contain less than 2 citation groups. For each seed in a citation group, we extract its perception vector from the surrounding sentences of that group. If the same citation group appeared multiple times in the paper, all surrounding sentences are used. The features of each instance (a cited paper) is the bag of words of its title and abstract. Finally, we get 547 clusters from 97 papers, 232 cited papers, 2227 data features, and 2465 perception features. Note that the perception vectors in these datasets may contain noise.

**Data Augmentation.** The above baselines, when used as standalone clustering algorithms, do *not* utilize the perception vectors by default. To achieve fair comparison, we *augment the datasets with perception features* for these standalone baselines so every algorithm considered in our experiments has an equal amount of information as input. The augmented input for user $u$ is $\boldsymbol{X}_u \in \mathbb{R}^{n \times (d+b_u)}$. Each instance has $d + b_u$ features. The last $b_u$ features are the perception features given by user $u$. Only the instances with seeds/constraints have non-zero values in the last $b_u$ features, which are the union of the perception features over the clusters which the respective instances belong to.

**General Results.** We first compare the recalls, precisions, and $F$-scores of the 12 algorithm combinations on real datasets. We randomly select 2 instances from each of the user-perceived cluster as seeds. The results are shown in Table 1. With Mturk image and song datasets where the perception features represent higher-level concepts (i.e., reasons and tags respectively), NPE is able to yield 35%~63% improvement in $F$-score when paired up with different baselines. The only exception is MOC* on Mturk dataset, with which NPE gives 10% improvement. We believe this is because that MOC*, a generative model, makes stronger assumption on feature distributions and, when the assumption holds, can give superior performance than discriminative methods. On the other hand, MOC* is less stable as its performance degrades when the assumption does not hold. We can see this by the drop on the song dataset. Preceding MOC* by NPE makes the performance of MOC* stable and gives up to 45% improvement in $F$-score (on song dataset).

With the citation dataset where the perception features and data features are at the same semantic level (i.e., words), *both* LPE *and* NPE are able to give 9%~14% improvement. In this case, the perception features and data features are likely to be linearly correlated. Therefore, LPE suffices to make good use of the perception features. Generally, LPE takes less time to train than NPE as there is no hyperparameter from the kernel function to tune. Note that the data features of all datasets are augmented with perception features so the standalone baselines have perception features as input. However, through its novel objective formulation, PE can make much better use of perception vectors—by turning them into the feature-level supervision.

**Back to the Case Study.** In Section 2, we have shown a case of large gap between the user-perceived clusters and data-driven clusters. We study how PE helps in this case. Figure 3(a) shows that, when taking perception features into account, both LPE and NPE capture user-perceived clusters more precisely, and there is 84% and 100% reduction in the number of users who get $F$-score lower than or equal to 0.5 respectively. Next, we study how PE copes with the sampling bias. Figure 3(b) shows the $F$-score histogram of the clusters found by LPE and NPE for a randomly picked user given different seed sets. Preceding OKM* by LPE and NPE respectively gives 94% and 100% reduction in the occurrence of data-driven clusters whose $F$-scores are lower than or equal to 0.5 due to "bad" seed sets. The sampling of seeds no longer plays a crucial role in performance. We observe similar advantages when pairing up PE with other baselines. See Section 4.2 of the supplementary.

Table 1: The recalls and precisions given by different algorithms on different real datasets.

| | Mturk (Image) | | | Song | | | Citation (Text) | | |
|---|---|---|---|---|---|---|---|---|---|
| | Recall | Prec. | $F$-score | Recall | Prec. | $F$-score | Recall | Prec. | $F$-score |
| OKM* | 0.461 | 0.523 | 0.490 | 0.575 | 0.358 | 0.441 | 0.889 | 0.713 | 0.792 |
| LPE+OKM* | 0.679 | 0.566 | 0.625 | 0.611 | 0.356 | 0.450 | **0.925** | 0.809 | 0.863 |
| NPE+OKM* | **0.782** | **0.694** | **0.735** | **0.771** | **0.664** | **0.713** | 0.920 | **0.856** | **0.887** |
| SEC* | 0.476 | 0.526 | 0.500 | 0.547 | 0.394 | 0.458 | 0.927 | 0.693 | 0.793 |
| LPE+SEC* | 0.754 | 0.646 | 0.696 | **0.739** | 0.371 | 0.494 | **0.951** | 0.898 | 0.925 |
| NPE+SEC* | **0.769** | **0.727** | **0.748** | 0.714 | **0.780** | **0.745** | 0.937 | **0.915** | **0.926** |
| CCSKL* | 0.429 | 0.441 | 0.435 | 0.404 | 0.353 | 0.377 | 0.890 | 0.667 | 0.762 |
| LPE+CCSKL* | **0.654** | 0.527 | 0.584 | 0.489 | 0.361 | 0.415 | **0.913** | 0.759 | **0.834** |
| NPE+CCSKL* | 0.642 | **0.545** | **0.589** | **0.627** | **0.449** | **0.525** | 0.899 | **0.766** | 0.827 |
| MOC* | 0.655 | 0.576 | 0.614 | 0.555 | 0.383 | 0.453 | 0.849 | 0.752 | 0.798 |
| LPE+MOC* | 0.610 | 0.621 | 0.615 | 0.646 | 0.393 | 0.489 | **0.901** | 0.856 | 0.878 |
| NPE+MOC* | **0.667** | **0.685** | **0.676** | **0.767** | **0.568** | **0.655** | 0.895 | **0.925** | **0.910** |

# 5 Conclusions

We demonstrate a gap between data-driven and user-perceived clusters due to the sampling bias, and propose LPE and NPE to bridge the gap by exploiting the feature level supervision in the form of perception vectors. To the best of our knowledge, this is the first study that proposes learning from the perception vectors to improve clustering and opens up numerous research directions. For example, it is important to investigate the theoretical performance guarantee and how the improvement is bounded when different numbers of perception vectors/features are available. It is also worth studying how different kernel functions affect the performance of NPE. These are matters of our future inquiry. To facilitate such exploration, we make our Mturk dataset public at http://cs.nthu.edu.tw/~shwu/datasets/shwu-mturk-16.zip.

## Footnotes

[1]The reason we choose an overlapping clustering algorithm is that from our experiments, the clusters perceived by a user can overlap. This observation is consistent with previous studies [1, 7]. During each

[4]Note that the traditional formula for precision should use $|C_{j^*}|$ as the denominator. Such precision may be very small since the number of instances picked by a user in $\mathcal{D}$ is usually much smaller than the number of all instances. And the precision would get smaller along with the increasing of the dataset size. Therefore, in order to get meaningful precision, we calculate the precision only based on the instances that the user picked.

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
