[Supplementary Material · user_preceived_clustering_13-nips-sup-cr2.pdf]

# Learning User Perceived Clusters with Feature-Level Supervision: Supplementary Materials

Ting-Yu Cheng   Kuan-Hua Lin   Xinyang Gong   Kang-Jun Liu   Shan-Hung Wu
{tycheng,khlin,xygong,kjliu}@datalab.cs.nthu.edu.tw   shwu@cs.nthu.edu.tw

## 1   Availability of Perception Vectors

It might not be feasible in some applications to ask users for their explicit perception as in our Mturk experiment. If so, the perception vectors can be obtained *implicitly*, just as how we get them from the DBLP citation dataset. For example, in Youtube or Spotify, a user may have maintained his/her own playlists. Each playlist has a title, which can be treated as the perception about why the user put certain videos/songs (i.e., seeds or must-linked instances) together in the playlist. So, to cluster videos/songs and align to that user's perception, one can extract the perception vectors from the titles. We can easily transfer this idea to other applications:

- To cluster images in Flickr: we can use the descriptions of photo albums as the perception vectors.

- To cluster users in Facebook: Facebook provides the "group" pages. Each group in Facebook has a name and description. The name and the description can be considered as the reasons why the group members joined the group. Thus, the name and the description of the Facebook groups can be the perception vectors we need to cluster users.

- To cluster web pages or products in a search engine: in Google or Amazon product searcher, the query typed by users can be the perception vectors of the seed set containing the webpages/products the users clicked among the search results.

There are lots of similar ways to extract the perception vectors in more applications.

## 2   Objective, Regularizers, and Unsmoothness

In Eq. (2) of the main text, the effect of the second term is covered by the third term, as two instances constrained by a must-link are associated to the same perception vectors (recall that a perception annotates a must link). So, requiring the latent representation of the two instances to be similar to the same perception vector (in the third term) implies requiring the latent representation of the two instances to be similar (in the second term). We obtain a simplified objective:

$$\arg\min_{F,W,b} \|XW + 1_n b^\top - F\|_F^2 + \beta \|S(F - P)\|_F^2, \tag{1}$$

This objective can further be regularized to increase performance. Generally, we prefer the linear embedding to as simple as possible to maximize the propagation effects. Also, since entries of $P$ are non-negative, we require $F$, $W$, and $b$ be non-negative as well. This leads to the final LPE objective, as shown by the Eq. (3) of the main text.

Note that, unlike most spectral clustering algorithms [1, 2, 4], we do *not* regularize $F$. This is because such regularization makes $F$ (so are the final clusters) generalizable to different users. In unsupervised clustering, this makes sense since the user perception is undefined, and the best policy is to maximize the generalizability to make everyone happy. However, in semi-supervised clustering whose goal is to approximate some unknown perception *of a particular user*, a better

Figure 1: Unsmoothness. (a) Histogram of overlapping scores when comparing two clusters having at least two common seeds. (b) Histogram of overlapping scores when comparing two clusters having identical perception features.

---

**Algorithm 1** Training algorithm for LPE.

---

    **Input:** data $\boldsymbol{X}$, seeds $\mathcal{S}$, perception features $\mathcal{P}$, and stopping criteria $\epsilon$;
    **Output:** $\mathcal{C}$;
    Initialize $\boldsymbol{F}$, $\boldsymbol{b}$, $\boldsymbol{W}$ randomly and set $t = 1$;
    **repeat**
        Update $\boldsymbol{F}$, $\boldsymbol{b}$, and $\boldsymbol{W}$ using Eqs. (2), (3), and (4) respectively
        $v^{(t+1)} \leftarrow \|\boldsymbol{X}^{\mathsf{T}}\boldsymbol{W} + \mathbf{1}_n\boldsymbol{b}^{\mathsf{T}} - \boldsymbol{F}\|_F^2 + \beta\|\boldsymbol{S}(\boldsymbol{F} - \boldsymbol{P})\|_F^2 + \gamma\|\boldsymbol{W}\|_F^2$;
        $t \leftarrow t + 1$;
    **until** $v^{(t)} - v^{(t+1)} < \epsilon$
    Compute $\mathcal{C}$ by applying an overlapping (seeded) clustering algorithm to $\boldsymbol{F}$.

---

policy would be to exploit the supervision of the user to maximize specificity. To justify this, we show an example using the Mturk image dataset collected from Section 2 of the main paper. For each pair of clusters $D_i$ and $D_j$ perceived by two different users $i$ and $j$, we calculate an overlapping score by $\mathrm{overlap}(D_i, D_j) = |D_i \cap D_j|/|D_i \cup D_j|$. Figures 1(a) and (b) show the histograms of overlapping scores for those pairs having at least two common seeds and identical perception features respectively. As we can see, two users may perceived very different clusters *even they give some common seeds or identical perception features*. In this case, regularizing $\boldsymbol{F}$ can actually hurt performance.

# 3 Training

**LPE.** To solve the non-negative $\boldsymbol{F}, \boldsymbol{b}$ and $\boldsymbol{W}$, we followed the multiplicative update rules proposed by Long et al. [3] to update $\boldsymbol{F}, \boldsymbol{b}$ and $\boldsymbol{W}$ iteratively. The update rules are given as follows:

$$\boldsymbol{F} \leftarrow \boldsymbol{F} \circ \sqrt{\frac{\boldsymbol{XW} + \mathbf{1}_n\boldsymbol{b}^{\top} + \beta\boldsymbol{SP}}{\boldsymbol{F} + \beta\boldsymbol{SF}}}, \tag{2}$$

$$\boldsymbol{b} \leftarrow \boldsymbol{b} \circ \sqrt{\frac{\boldsymbol{F}^{\top}\mathbf{1}_n}{\boldsymbol{W}^{\top}\boldsymbol{X}^{\top}\mathbf{1}_n + \boldsymbol{b}\mathbf{1}_n^{\top}\mathbf{1}_n}}, \tag{3}$$

$$\boldsymbol{W} \leftarrow \boldsymbol{W} \circ \sqrt{\frac{\boldsymbol{X}^{\top}\boldsymbol{F} + \gamma\boldsymbol{W}}{\boldsymbol{X}^{\top}\boldsymbol{XW} + \boldsymbol{X}^{\top}\mathbf{1}_n\boldsymbol{b}^{\top} + \alpha\boldsymbol{W} + \gamma\boldsymbol{WW}^{\top}\boldsymbol{W}}}, \tag{4}$$

where operator $\circ$ denotes element-wise product, $\frac{[\cdot]}{[\cdot]}$ denotes the elemental-wise division, and $\sqrt{\cdot}$ denotes elemental-wise square root. And the training steps are shown in Algorithm 1.

**NPE.** To solve the non-negative $\boldsymbol{F}$ and $\boldsymbol{C}$, we first observe that both $\boldsymbol{K}$ and $\boldsymbol{F}$ have non-negative elements. So, to be efficient, we narrow the search space of $\boldsymbol{C}$ down to the positive matrices. We then apply the multiplicative update rules [3] to update $\boldsymbol{F}$ and $\boldsymbol{C}$ iteratively. The update rules are given as follows:

$$\boldsymbol{F} \leftarrow \boldsymbol{F} \circ \sqrt{\frac{\boldsymbol{KC} + \beta\boldsymbol{SP}}{\boldsymbol{F} + \beta\boldsymbol{SF}}}, \tag{5}$$

---

**Algorithm 2** Training algorithm for NPE.

---
**Input:** data $\boldsymbol{X}$, seeds $\mathcal{S}$, perception features $\mathcal{P}$, kernel function $k$, and stopping criteria $\epsilon$;
**Output:** $\mathcal{C}$;
Initialize $\boldsymbol{F}$, $\boldsymbol{C}$ randomly and set $t = 1$;
Calculate the kernel matrix $\boldsymbol{K}$ of $\boldsymbol{X}$ based on $k$;
**repeat**
    Update $\boldsymbol{F}$ and $\boldsymbol{C}$ using Eqs. (5) and (6) respectively
    $v^{(t+1)} \leftarrow \|\boldsymbol{KC} - \boldsymbol{F}\|_F^2 + \beta\,\|\boldsymbol{S}(\boldsymbol{F} - \boldsymbol{P})\|_F^2 + \gamma \operatorname{tr}(\boldsymbol{C}^\mathsf{T} \boldsymbol{KC})$;
    $t \leftarrow t + 1$;
**until** $v^{(t)} - v^{(t+1)} < \epsilon$
Compute $\mathcal{C}$ by applying an overlapping (seeded) clustering algorithm to $\boldsymbol{F}$.

---

Table 1: Statistics of the user-perceived clusters (ground truth) of the Mturk image dataset, song dataset, and citation dataset.

|  | Mturk (Image) | | | Song | | | Citation (Text) | | |
|---|---|---|---|---|---|---|---|---|---|
|  | Min | Max | Avg. | Min | Max | Avg. | Min | Max | Avg. |
| Number of clusters per user | 2 | 7 | 3.3 | 3 | 3 | 3 | 2 | 12 | 4.21 |
| Cluster size | 2 | 128 | 21.8 | 10 | 181 | 36 | 2 | 23 | 4.25 |
| Number of clusters per instance | 5 | 91 | 39.3 | 1 | 10 | 2 | 1 | 27 | 2.35 |

$$C \leftarrow C \circ \sqrt{\frac{KF}{KKC + \gamma KC}}. \tag{6}$$

The training steps are shown in Algorithm 2.

**Producing Final Clusters.** After obtaining $\boldsymbol{F}$, PE can then paired with any existing unsupervised or semi-supervised clustering algorithms to produce the final clusters $\mathcal{C}$'s. Since the user-perceived clusters are observed to be overlapping (see Section 2 of the main paper), we focus on the overlapping clustering algorithms. Furthermore, although ideally a seeded semi-supervised clustering algorithm should be adopted to ensure the one-to-one correspondence between the seed sets $\mathcal{S}_j$'s and the found clusters $\mathcal{C}_j$'s, we observe that in most case, an unsupervised clustering algorithm can also lead to the one-to-one correspondence.

## 4    More on Experiments

### 4.1    Details of Settings

For parameter tuning, in SEC* and CCSKL* we need to specify an affinity matrix $\boldsymbol{A}$ and tune $\sigma$ to obtain a proper distance metric $\exp(-\frac{\|x_i - x_j\|^2}{\sigma^2})$. We follow the self-tuning method proposed by [5]. For hyperparameters $\mu$ and $\gamma$ in SEC*, we consider $\mu$ from $\{10^{-2}, 10^0, 10^2\}$, $\gamma$ from $\{10^{-4}, 10^{-2}, 10^0, 10^2, 10^4\}$, and report the best result form all combinations. In CCSKL*, $m$, the number of eigenvectors preserved, needed to be determined. We set $m = 20$ by following settings in the original paper. However, when paired up with LPE, the data dimension is mapped into a lower dimension, $k$. Therefore, we set $m = k$ in this case. We give the hyperparameter $\beta$ in LPE a sufficiently large number so that the final clusters align with the seeds.

All of the algorithms considered in this paper have non-convex objectives. So their performance depend on the initialization points in the training process.[1] For each reported performance, we repeat the training process 10 times with random initialization and record the result with the lowest objective score.

Table 1 summarizes some statistics of the ground truth clusters in our real datasets.

Figure 2: The $F$-Score histogram of (a) SEC* alone, LPE+SEC*, and NPE+SEC*; (b) CCSKL* alone, LPE+CCSKL*, and NPE+CCSKL*; (c) MOC*, LPE+MOC*, and NPE+MOC* applied to the Mturk dataset.

## 4.2 Recall Histograms

Continuing the discussion in Section 4 of the main text, , we study the recall histograms given by various algorithms, which are shown in Figure 2. In most cases, the low quality data-driven clusters (whose $F$-scores are lower than or equal to 0.3) due to "bad" seeds can be eliminated by LPE and NPE. The only exception is LPE+MOC*, which gives relatively little improvement. We believe this is because of the strong assumption about the feature distribution employed by MOC*. Recall that MOC* is a generative model and its assumption seems to hold in the Mturk dataset (regarding data features). The feature-level supervision may conflict with such an assumption, but not strong enough to change the behavior of MOC*.

## Footnotes

[1]The only exception is OKM*, which initializes at seeds.