[Reviews · NeurIPS 2016]

Reviewer 1

Summary

The authors propose a solution to integrate a particular type of side information (called here perception vectors) into clustering algorithms as a new way to implement user supervision. More precisely, classical user supervision such as seeds or link/do not link constraints are supposed to be complemented by the so-called perception vectors (e.g. a vector space representation of a text justifying the constraints). Those vectors are not described in the same vector space as the original data. The authors propose to embedded the original data into the perception vector space in a way that respects the specified constraints. Classical clustering algorithms are applied to the embedded data.

Qualitative Assessment

This paper is clearly interesting but it suffers from being squeezed into the page limit of NIPS. In my opinion, the paper does not stand alone (without the supplementary materials) for several reasons. Firstly, reproducing the results without the supplementary materials (SM) is probably very difficult (the algorithms and all the technical details are only given in the SM). Secondly, I had to refer to the SM to get a good grasp at the perception vector concept (more on that latter). Moreover, the experiments contained in the main paper are a bit limited. Finally, even with all that has been transfered to the SM, the paper does not contain any conclusion and real discussion on the results. Apart from this obvious problem (it's not really a flaw of the paper but rather a consequence of the page limit of the conference), I also think that the main "selling point" of the paper, that is the sampling bias, is not very convincing (I'm convinced by the results, not much by the explanations) or at least incorrectly presented. Indeed, the authors only show that their method is less sensitive to the actual constraints provided by the users when they make use of perception vectors, while classical algorithms can be mislead by incorrect or inappropriate constraints. The problem is that users can be "tricked" into giving wrong feedback (for instance when shown ambiguous pictures without much context) rather than a sampling problem. In addition, and this is explained quite well in the paper, users base their reasoning on some external assumptions and on additional features that are not accessible to standard algorithms. So the key point here appears to be the possibility of integrating additional features rather than avoiding some possible (but not shown) sampling bias. In order to fix those problems, I would recommend the authors to reduce the discussion on the sampling bias to a bare minimum (maybe move it to the SM) so as to bring back in the main paper the much needed technical details. There is also some redundancy in the paper as the main solution (seeded OKM + the proposed embedding) is described twice (page 3 and page 6).

Confidence in this Review

2-Confident (read it all; understood it all reasonably well)


Reviewer 2

Summary

This paper discusses a special semi-supervised clustering algorithm that considers feature-level constraints via user input. It claims it goes beyond instance-level supervision, and is using so-called perception embedding to learn the mapping. The paper provides good motivation to the feature-level supervision, but I have a few doubts on how this can be applied in real-world applications. The users would normally give high-level supervision instead of at the feature level. What is the exact supervision the Turkers gave in the MTurk example shown in this paper? My understanding is that the users provide certain grouping of images via keywords, which is not what I'd call the feature level. It's more likely that the users provide certain grouping which can be translated into must-link or cannot-link relationships. Also I don't understand what's the "perception features" for the song and citation data sets that are used in experiments. From the formulation (1), it seems the latent variable matrix F and the perception matrix P has the same number of columns (i.e. dimensions). Is this a coincidence or a requirement for the proposed approach? From what I understand the perception can be in any dimensions, which may or may not be the same as the latent dimensionality. I also don't understand why the instance-level constraints can be dropped from the formulation in (3). What's the time and space complexity of the proposed algorithm? It seems all the 3 test data sets are fairly small so I'm not sure how large-scale can the proposed approach be used. How sensitive are the parameters (dimensions, kernel parameters, etc) to the final results? I'm asking this because there are quite a few parameters to tune so we need to make sure the results are robust. A minor comment is that the authors put a lot of details into supplementary materials. They should have shortened much of the motivation and discussions, and put those important details in, e.g. algorithms, complexity, data summary.

Qualitative Assessment

I would suggest the authors to clarify the motivation better, maybe starting with a more clear image for figure 2. Throughout the paper I'm not very clear on what the perception features are. Also the paper can be restructured to have more key details with less lengthy discussions/details.

Confidence in this Review

2-Confident (read it all; understood it all reasonably well)


Reviewer 3

Summary

This paper proposes a novel clustering algorithm which utilizes the side information to consider the user perceptions. The basic idea is to introduce a so called perception embedded vector to identify the key features that affect the user perceptions.

Qualitative Assessment

The idea is interesting and reasonable. It can reduce the negative affect of the sample bias problem in semi-supervised clustering methods by utilizing the feature level supervision. Experiments are performed on several datasets to compare with some baselines, and results validated the effectiveness of the proposed method. I am curious about how would the proposed method work if the constrains come from different users. Because different users may cluster the data based on different features. And then it would be less possible to learn a unified perception embedding. I would like to see the affection of the parameter \beta in Equation 3. An empirical study on the sensitivity of parameters can better show how the PE works. I am not sure why overlapping clustering is a necessary? This constraint makes the baselines weak in the experiments. It is more common to perform non-overlapping clustering. It would be interesting to also report the results of baselines with only the original features. The datasets used is quite small with only 180 images. Figure 2 is less readable; authors may want to replace it with a clear one.

Confidence in this Review

2-Confident (read it all; understood it all reasonably well)


Reviewer 4

Summary

The abstract describes it quite well. In the process of semi-supervised-clustering, a user can add contextual information to each cluster seed. A linear (&optionally nonlinear) embedding from the original features to this "perceptual" space is then learned. Once embedded, clustering is done & performance improved.

Qualitative Assessment

The approach/problem seemed novel/valuable to me, and the technique sane and technically justified. Linear/Nonlinear PE objective formulations seem reasonable. The section on "Avoiding Sampling Bias" made a clear point, but the title & claim that this "avoids sampling bias" seems like a different claim. The concrete claim made in that section is: the linear PE objective results in perceptual feature similarity, as well as hard-seed-constraint-similarity being accounted for. While capturing additional notions of similarity in the objective appears to boost performance, I don't see how this is explicitly aiming at correcting sampling bias. Sampling bias corrections usual involve some model of sampling, and some post-correction based on this model.

Confidence in this Review

2-Confident (read it all; understood it all reasonably well)


Reviewer 5

Summary

This paper describes a semi-supervised clustering technique--an extension of spectral embedded clustering--whereby users provide low-dimensional labels called “perception embeddings” for a few seed data points in each cluster (ex: user provides a few words describing each of several seed images per cluster, in an image clustering task), and these are used to help learn a low-dimensional embedding of the data points to cluster them. In particular, these perception embeddings do not need to be in the same feature space as that of the data points. Crucially, this method focuses on cases where there are only a small percentage of data points labeled as seeds, and where there may be sampling bias in selecting the seeds. Noteworthy empirical gains are presented on a range of datasets.

Qualitative Assessment

The empirical results seem impressive and thorough. Overall, this seems to be a simple yet powerful approach that I could see being of great practical use in many real world scenarios- the perception embeddings seem easy to describe in many different useful cases, and the core formulation of the learning objective is a simple and clear extension of popular spectral methods. The main weakness of the paper, in my opinion, is the clarity of the technical explanation of why the method works over just selecting seeds i.e. specifying data points in feature space, specifically, the explanation of why the method is more tolerant to “misleading” seed sets. This is explained well initially at a high level in the intro, but somewhat poorly and vaguely in the “avoiding sampling bias” section. Theoretical guarantees, or something slightly more formal / rigorous / quantitative defining how the method avoids sampling bias / tolerates variance across seed sets, and “bad” seed sets in particular, would be very helpful here. For example, even just defining a metric for measuring the bias of a seed set (and therefore what exactly a “bad”/”misleading” seed set is) would have been helpful to start. Perhaps some of the material from the appendix could be helpful here as well. One question in particular comes to mind for me: how are the perception embeddings actually more helpful than just an indicator vector of class labels for the seed points? In Fig. 4(b), the use of lower-dimensional PEs is explored, but it doesn’t specifically answer this question. There is a nice toy example in Appendix Fig. 3, however here the PEs are the class labels. This seems like an important question to address? A few minor points: - Lines 121-2: What does “skipping spoliers” and dropping “low quality” clusters mean? - Line 226: Where is Theorem 1? - Editing for grammar / typos would be useful, esp. In the “avoiding sampling bias” section (lines 174-207), and lines 272-279 for example - The point that this method is particularly applicable when there is a small amount of supervision / seeds specified would be useful to emphasize more upfront

Confidence in this Review

2-Confident (read it all; understood it all reasonably well)


Reviewer 6

Summary

The authors present a framework for Perception Embedded clustering, which combines side information from the user and traditional clustering information to create a semi-supervised clustering. The key insight is to use information provided by the user at the feature-level to identify the data features that the user finds useful ("cares about") in clustering. The introduce two versions of PE, linear and non-linear PE, discuss some theoretical properties non-linear PE, and demonstrate the efficacy of the method on a number of common clustering datasets.

Qualitative Assessment

This work has potential to be very valuable to the field given the empirical results the authors show. There are two main issues with the paper as far as I can tell: 1. There is far too much time spent in sections 1 and 2, detailing the problem and perception gap. I didn't really grasp what the authors' contribution was until section 3. This should be better explained in the abstract and in the introduction. That is, the authors should state the general idea of their PE approach much earlier in the text, e.g. in the abstract and/or introduction. The authors should give some intuition as to why they take this approach, and why their approach will lead to better clustering results than the common methods to which they compare. 2. There are seemingly important empirical results buried in the supplemental material, when these could have a strong positive impact on how the paper is received. For example, Figure 5a shows that the run-time of the non-linear PE algorithm is roughly constant in the number of data features, which is very important. However, this is only discussed briefly. There are other nice results and explanations of the experiments in the supplementary materials, leaving me to wonder if NIPS is really the most appropriate destination for this work. The paper is generally well-written. Here is a non-exhaustive list of typos: "pairwise constrains"; "if the user experience difficulty"; "When clustering musics/videos"; "Table 1 in supplementary."; missing a comma in "nonlinear correlations respectively"; "which" should be "that" in "data-features which the user really cares about";

Confidence in this Review

1-Less confident (might not have understood significant parts)